# Successful Use of Methylene Blue in Catecholamine-Resistant Septic Shock: A Case Report and Short Literature Review

**DOI:** 10.3390/ijms241310772

**Published:** 2023-06-28

**Authors:** Michał P. Pluta, Zbigniew Putowski, Piotr F. Czempik, Łukasz J. Krzych

**Affiliations:** 1Department of Anaesthesiology and Intensive Care, Faculty of Medical Sciences in Katowice, Medical University of Silesia, 40-752 Katowice, Poland; anestezjologia.ligota@sum.edu.pl (Z.P.); pczempik@sum.edu.pl (P.F.C.); lkrzych@sum.edu.pl (Ł.J.K.); 2Department of Cardiac Anaesthesia and Intensive Therapy, Silesian Center for Heart Diseases, Medical University of Silesia, 41-808 Zabrze, Poland

**Keywords:** vasoplegic shock, sepsis, methylene blue

## Abstract

Despite efforts to improve treatment outcomes, mortality in septic shock remains high. In some patients, despite the use of several adrenergic drugs, features of refractory vasoplegic shock with progressive multiorgan failure are observed. We present a case report of the successful reversal of vasoplegic shock following the use of methylene blue, a selective inhibitor of the inducible form of nitric oxide synthase, which prevents vasodilation in response to inflammatory cytokines. We also briefly review the literature.

## 1. Case Report

A 67-year-old male patient with septic shock due to right-sided pneumonia with parapneumonic effusion was admitted to the Intensive Care Unit (ICU). The patient’s medical history included: persistent atrial fibrillation, abdominal aortic aneurysm, previous toxic liver injury, chronic kidney disease, type 2 diabetes, and malnutrition.

On admission, the patient was conscious. He required an intensive oxygen therapy via a non-rebreather oxygen mask (15 L per min) to achieve SpO_2_ > 90%. The APACHE II (Acute Physiology and Chronic Health Evaluation), SAPS (Simplified Acute Physiology Score), and SOFA (Sequential Organ Failure Assessment) scores were 18, 31 and 8 points, respectively. In the following hours, the patient’s respiratory support was escalated to non-invasive mechanical ventilation, followed by endotracheal intubation and invasive mechanical ventilation. The patient was managed according to the most recent Surviving Sepsis Campaign guidelines [1]. The patient was started on empiric broad-spectrum antibiotic therapy (meropenem, linezolid) and was resuscitated with a balanced crystalloid (Optilyte, Fresenius Kabi, Bad Homburg, Germany). To achieve optimal mean arterial pressure (MAP), the following infusions were started sequentially and their infusions increased gradually to maximal doses of: norepinephrine (NE)—2.2 µg kg^−1^ min^−1^, epinephrine (E)—0.2 µg kg^−1^ min^−1^, arginine vasopressin (AVP)—0.04 IU min^−1^. The patient was also started on intravenous corticosteroids (50 mg hydrocortisone succinate QID). Transthoracic echocardiography revealed hyperkinesis of both ventricles, and no other pathology was found. Despite the implemented therapy, the patient developed severe metabolic acidosis (pH 7.03, BE^−15.9^ mmol L^−1^) and acute kidney injury with anuria and hyperkalaemia. Consequently, the patient was started on continuous veno-venous hemodiafiltration with regional citrate anticoagulation. Due to spontaneous hypoglycaemia the patients was started on continuous infusion of 40% dextrose. Due to the refractory nature of the shock, methylene blue (MTB, Proveblue, Provepharm, Marseille, France) was started as a rescue therapy—initial bolus (1 mg kg^−1^) as a 10-min infusion followed by 6-h infusion (0.5 mg kg^−1^ h^−1^). As quickly as an hour after MTB initiation, doses of catecholamines were reduced significantly. After 6 h, E infusion was terminated and the dose of NE was reduced to 0.5 µg kg^−1^ min^−1^. After 9 h from MTB initiation, the infusion of AVP was ceased and NE was maintained at a dose of 0.3 µg kg^−1^ min^−1^ with preserved MAP > 65 mmHg and normal systemic vascular resistance (SVR). Gradual normalization of the acid–base balance parameters was achieved (pH 7.38, HCO_3_ 22.3, base deficiency/excess −4.1). On the next day, NE infusion was reduced to a dose of 0.04 µg kg^−1^ min^−1^ (Figure 1). The culture of blood and bronchoalveolar lavage revealed *Streptococcus pneumoniae* infection. On day 5 the catecholamine infusion was stopped, and on day 6 the patient was weaned off the respirator and extubated. The patient remained conscious and made logical contact with the ICU staff.

## 2. Discussion

Despite efforts to improve treatment outcomes, mortality in septic shock remains high [1]. Along with the localization of the source of infection, the use of empiric antibiotic therapy and liberal fluid therapy, the prompt initiation of alpha-adrenergic agents is recommended to restore and maintain normal SVR [1]. With hemodynamic coherence preserved, normalization of macrocirculatory parameters increases the likelihood of improved microcirculatory perfusion [2]. However, in some patients, despite the use of several adrenergic drugs (norepinephrine, epinephrine, phenylephrine), features of refractory vasoplegic shock (VS) with progressive multiorgan failure are observed [3]. Since the large doses of catecholamines required to reverse vs. induce numerous side effects (myocardial stunning, tachycardia, hypercoagulability, adverse immunomodulation and visceral compartment hypoperfusion with bacterial translocation), catecholamine-sparing therapies are being investigated [4].

The concept of using MTB in vs. has a pathophysiological rationale. Nitric oxide synthase (NOS) in the vascular endothelium converts L-arginine to L-citrulline and nitric oxide (NO), which, due to its small size, diffuses freely across cell membranes, enters smooth muscle cells and causes dephosphorylation of GTP. The produced cGMP blocks the release of calcium from the sarcoplasmic reticulum, causing smooth muscle relaxation [5]. Under septic shock conditions, bacterial endotoxins and numerous proinflammatory cytokines promote cGMP production by increasing the expression of the inducible form of NOS (iNOS). This increases available NO stores until iNOS reserves are completely depleted. Excessive NO production leads to a decrease in SVR with some patients developing vs. [5] (Figure 2).

In addition to generating unfavorable vasoplegia, NO also mediates the inhibition of thrombocyte aggregation, counteracts oxidative stress and prevents cell apoptosis; therefore, the use of non-selective inhibitors of the NO-cGMP pathway that block all isoforms of NOS (neuronal, endothelial and inducible) is harmful, as shown in previous studies [5]. On the contrary, MTB, as a selective iNOS inhibitor, may be a potential therapeutic option in vs. that preserves the beneficial profile of NO [6].

The lack of recommendations for the routine use of MTB is due to the lack of good-quality evidence, but the results of the small studies and case reports published to date confirm that some patients may respond positively to MTB.

The optimal dose of MTB needed to reverse vs. is unknown. In a prospective, randomized, placebo-controlled, double-blind study, Memis D et al. assigned 15 patients each to a group receiving MTB (0.5 mg kg^−1^ h^−1^) for 6 h in 100 mL 0.9%NaCl or placebo (100 mL 0.9% NaCl). Although there was no difference in mortality and inflammatory cytokine levels, MTB administration resulted in a significant increase in MAP (85 vs. 74 mmHg; *p* < 0.05). A major limitation of the study was the lack of information on the dosage of catecholamines in both groups before and after implementation of the study protocol [7]. Kirov, M.Y. et al. started treatment with a bolus of MTB at a dose of 2 mg kg^−1^ if norepinephrine and epinephrine requirements exceeded 0.05 μg kg^−1^ min^−1^, and then continued continuous infusion of MTB with a gradually increasing infusion rate starting at 0.25 mg kg^−1^ h^−1^ and doubling the dose every hour to a maximum dose of 2 mg kg^−1^ h^−1^. This treatment reduced the need for norepinephrine and epinephrine by more than 80% compared to placebo, while maintaining higher MAP. Survival in the MTB group was 50% compared to 30% in the group receiving conventional treatment [8]. Park et al. used MTB at a dose of 1 mg kg^−1^ in a 15-min infusion, observing a significant increase in MAP and SVR. Of the 11 patients who responded to MTB, 55% survived. In the group not responding to a single dose of MTB, mortality was 88% [9]. Similarly, in another study, MTB used at a dose of 1 mg kg^−1^ in a 15-min infusion in patients in septic shock who could not maintain adequate BP despite two vasopressors (NE > 0.2 μg kg^−1^ min^−1^ and E > 0.1 μg kg^−1^ min^−1^) resulted in a significant increase in MAP. There was no adverse effect of MTB on pulmonary function. However, despite the improvement in hemodynamic parameters, 8 out of 10 patients died [10]. The largest meta-analysis of 15 studies involving 832 vs. patients (including those in the setting of septic shock) showed that MTB given together with vasopressors reduced the odds of mortality by 46% (OR 0.54; 95%CI 0.34–0.85; *p* = 0.008) without serious adverse effects. In terms of hemodynamic parameters, MTB increased MAP, HR and SVR. It also showed a lower incidence of renal injury [11].

The optimal timing for initiating MTB therapy remains uncertain. Previous studies have used MTB as a rescue treatment when standard therapies fail to produce a response. This may have resulted in patients treated with MTB having worse metabolic parameters or being in the irreversible phase of shock, where the improvement of macrohemodynamic parameters was not able to restore adequate flow in the microcirculation (no-reflow phenomenon). In our case, MTB was successfully applied after more than 12 h of standard therapy, with a high demand for three vasopressors. Conversely, in an experimental model of lipopolysaccharide-induced sepsis (LPS), researchers hypothesized a time-dependent guanylate cyclase (GC) activity. Initial observations revealed increased iNOS activity during the first 8 h, followed by the downregulation of iNOS and decreased GC synthase activity in subsequent hours [12]. These findings suggest that the most opportune time to initiate MTB therapy is during the early stages of sepsis. In the latest study conducted by Ibarra-Estrada et al., the aim was to determine if the early inclusion of MTB would reduce the need for vasopressor support, leading to its complete discontinuation within 48 h [13]. Patients in the study received standard treatment as recommended by the Surviving Sepsis Campaign, with the intervention group additionally receiving a daily intravenous infusion of 100 mg MTB for 6 h, repeated three times. A placebo was administered to the control group. The use of MTB resulted in a shorter duration of vasopressor use (69 h, IQR 59–83) compared to the control group (94 h, IQR 74–141). In the MTB group, only 11% of patients required reinitiating NE infusion within 48 h of discontinuation, compared to 28% in the placebo group. However, mortality rates were similar between the two groups, and no serious adverse events were reported among patients receiving MTB.

### 2.1. Potential Toxicity and Side Effects

Data on MTB toxicity come mainly from case reports and small observational studies, and include doses much higher than those currently proposed for the treatment of VS. In several studies, MTB infusion at doses exceeding 3 mg kg^−1^ resulted in a significant increase in pulmonary vascular resistance and impaired gas exchange, suggesting caution in the use of MTB in primary pulmonary hypertension or pulmonary hypertension secondary to severe lung pathology [14]. We do not have reliable data on the safety of MTB use in patients with liver failure, in which the drug is metabolized, and renal failure is the organ responsible for MTB elimination [15]. In our case, we used MTB in a patient undergoing continuous renal replacement therapy (CRRT). Despite a discrete change in the staining of the filtrate, we did not observe any failure of the CRRT device, and the kidneys regained normal function at a later day. At higher doses of MTB, a transient change in skin coloration interfering with pulse oximetry readings is possible [16]. In cases of glucose-6-phosphate dehydrogenase deficiency, there is a risk of impaired MTB elimination, leading to the development of hemolytic anemia [17]. Since MTB is a reversible inhibitor of monoamine oxidase A, impaired metabolism of 5-hydroxytryptamine combined with chronic intake of serotonin reuptake inhibitors increases the risk of developing serotonin syndrome [18,19]. There are also reports of intrauterine fetal death after MTB use in the second trimester of pregnancy [15]. At MTB doses in the 1–2 mg kg^−1^ range, side effects are rare and take the form of dizziness, nausea or confusion, which is rarely a problem in the critically ill patient population due to pharmacologically abolished or reduced consciousness [15].

### 2.2. Research Limitations and Gaps

There is an evidence gap in the proper identification of patients susceptible to MTB treatment. From a pathophysiological point of view, MTB should be used if vasoplegia predominates and the cardiac index is normal or increased. But reliable assessments of vasoplegia alone in the clinical setting are limited. The SVR calculated in hemodynamic measurements shows estimated global vascular resistances, and does not take into account possible differences between the microcirculation of each organ dependent on local autoregulation. This creates a risk of local critical ischemia after MTB due to excessive vasoconstriction where vascular resistances were not initially reduced in this place.

In addition, septic shock often presents with concomitant myocardial dysfunction, resulting in decreased ejection fraction and, subsequently, cardiac index. In this group of patients, an excessive increase in afterload due to the vasoconstrictive effects of MTB may not be beneficial, although in macroparameters we can temporarily observe an improvement in MAP [20]. Although several studies on more numerous patient groups have been published to date, most have focused on achieving an adequate MAP above 65 mmHg without considering the mechanism of hypotension. In fact, most often, hypotension is due to vasoplegia, but this requires a more thorough clinical evaluation to exclude patients in whom MTB may be harmful. Our case demonstrates a personalized approach to the management of MTB therapy. MTB was given after the adequate filling of the vascular bed and the confirmation of normal cardiac index and normal cardiac contractility on echocardiography.

In our recent systematic review with meta-analysis, we showed that the normalization of MAP above the recommended 65 mmHg does not equate to improved capillary return time, confirming that improved microcirculatory conditions (not only MAP values) are a prerequisite for favorable resuscitation outcomes in septic shock [21]. MTB therapy has not been evaluated in the context of microcirculation, although we now have instruments that provide insight into microcirculation (e.g., Side-stream Darkfield Imaging). This area is definitely worth exploring.

## 3. Conclusions

The presented case shows that, although not yet confirmed by good-quality evidence, methylene blue may have a role as a rescue therapy in refractory septic shock.

## Figures and Tables

**Figure 1 ijms-24-10772-f001:**
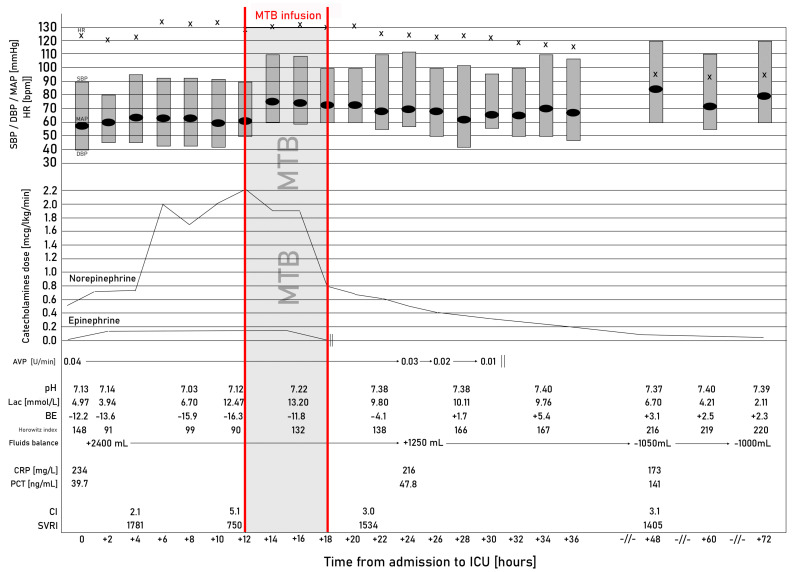
Selected clinical and laboratory parameters at subsequent time point (SBP, systolic blood pressure; DBP, diastolic blood pressure; MAP, mean arterial pressure; HR, heart rate; AVP, arginine vasopressin; Lac, lactates; BE, base excess; CRP, C-reactive protein; PCT, procalcitonin; CI, cardiac index; SVRI, systemic vascular resistance index; MTB, methylene blue).

**Figure 2 ijms-24-10772-f002:**
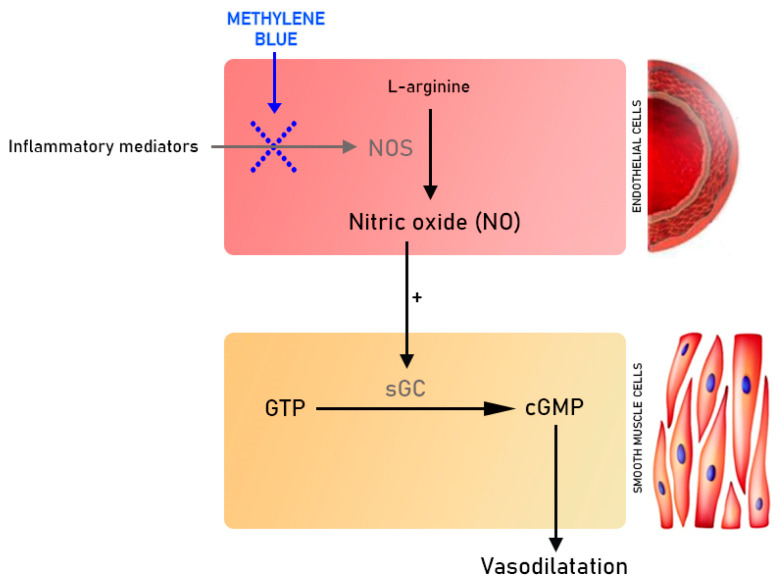
Mechanism of action of methylene blue (NOS, nitric oxide synthase; GTP, guanosine triphosphate; cGMP, nitric oxide-cyclic guanosine monophosphate; sGC, soluble guanylate cyclase).

## Data Availability

Data are available directly from the authors.

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
