# Peer review of "Successful Use of Methylene Blue in Catecholamine-Resistant Septic Shock: A Case Report and Short Literature Review"

_ijms, 2023, doi:10.3390/ijms241310772_

Round 1
Reviewer 1 Report
Pluta et al. provide a clinical case report addressing the use of methylene blue as rescue therapy in a septic shock patient. There was a significative response, with weaning of all vasopressors and the patient was discharged alive.
Clearly, septic shock mortality remains exceedingly high, and we need different strategies to address our patients.
1- The authors provide some references with far more patients receiving methylene blue (references 8-11), including one meta-analysis with 832 patients. Why do they think this report will add to the present knowledge?
2- Despite all therapeutic rational, methylene blue only provides an effective benefit in some patients. How do authors propose to address this issue? Is there any specific marker that may be used, or do they just suggest trying it and seeing what happens? And, if that is the case, for how long?
3- The authors extensively described the potential benefits of methylene blue in septic shock. However, I found the description of the potential toxicity poor. A balanced vision would be preferable.
Minor comments:
1- In Table lactate units should be provided. I am assuming it is mmol/dL.
2- I think authors are a little over-enthusiastic with methylene blue. Although, I think it might have a role in a few patients, overall, a clear mortality benefit is yet to be demonstrated. A more balanced view should be used.
I think it only required minor editing
Author Response
Thank you for taking the time to adjudicate our manuscript and for your important comments.
“1- The authors provide some references with far more patients receiving methylene blue (references 8-11), including one meta-analysis with 832 patients. Why do they think this report will add to the present knowledge?”
In fact, a few papers on a definite groups of patients have already been published, where the effect of MTB was evaluated. However, in our opinion, the design of these studies was too generalized - these studies usually assumed one of two scenarios:
(1) if the target mean arterial pressure >65mmHg was not achieved, MTB was included as rescue therapy
(2) MTB was used quite early, when the dose of norepinephrine exceeded the target value and MAP>65 was not achieved.
In fact, most often hypotension in septic patients is due to vasoplegia. However, there is a group of patients with concomitant impairment of cardiac contractility. In this group of patients, excessive vasoconstriction can be harmful. Currently, more and more attention is being paid to personalized medicine, i.e. tailored to the individual needs of the patient. Our case report highlights such an individualized approach. Our patient had refractory vasoplegia despite very high doses of pressor amines, but his heart still had preserved contractility. From a pathophysiological point of view, he was a patient who was likely to respond to MTB administration, and he did.
Our case also contradicts the replicated view that MTB applied after 8-12h of refractory shock has a low probability of improvement. Our patient received supramaximal doses of catecholamines for a short period of time, during which fluid resuscitation was performed, and then a catecholamine-sparing drug was applied after 12h with good results.
“2- Despite all therapeutic rational, methylene blue only provides an effective benefit in some patients. How do authors propose to address this issue? Is there any specific marker that may be used, or do they just suggest trying it and seeing what happens? And, if that is the case, for how long?”
We decided to clarify this doubt in an additional paragraph on research limitations and gaps:
There is an evidence gap in the proper identification of patients susceptible to MTB treatment. From a pathophysiological point of view, MTB should be used if vasoplegia predominates and the cardiac index is normal or increased. But assessment of vasoplegia alone in the clinical setting is limited. The SVR calculated in hemodynamic measurements shows estimated global vascular resistances and do not take into account possible differences between the microcirculation of each organ dependent on local autoregulation. This creates a risk of local critical ischemia after MTB due to excessive vasoconstriction where vascular resistances were not initially reduced in this place. In addition, septic shock often presents with concomitant myocardial dysfunction, resulting in decreased ejection fraction and cardiac index. In this group of patients, an excessive increase in afterload due to the vasoconstrictive effects of MTB may not be beneficial although in macroparameters we can temporarily observe an improvement in MAP. Although several studies on more numerous patient groups have been published to date, most have focused on achieving an adequate MAP above 65 mmHg without considering the mechanism of hypotension. In fact, most often hypotension is due to vasoplegia, but this requires a more thorough clinical evaluation to exclude patients in whom MTB may be harmful. Our case demonstrates a personalized approach to the management of MTB therapy. MTB was given after adequate filling of the vascular bed and confirmation of normal cardiac index and normal cardiac contractility on echocardiography.
In our recent systematic review with meta-analysis, we showed that normalization of MAP above the recommended 65mmHg does not equate to improved capillary return time, confirming that improved microcirculatory conditions (not just only MAP values) are a prerequisite for favorable resuscitation outcome in septic shock. MTB therapy has not been evaluated in the context of microcirculation, although we now have instruments that provide insight into microcirculation (e.g. Side-stream Darkfield Imaging). This area is definitely worth exploring.
“3- The authors extensively described the potential benefits of methylene blue in septic shock. However, I found the description of the potential toxicity poor. A balanced vision would be preferable.”
We have included an additional paragraph addressing the aspect of side effects and toxicity, although we would like to note that severe adverse events have not typically been reported at doses around 1-2 mg/kg body weight.
“I think authors are a little over-enthusiastic with methylene blue. Although, I think it might have a role in a few patients, overall, a clear mortality benefit is yet to be demonstrated. A more balanced view should be used.”
Demonstrating that any pharmacological intervention in the ICU patient population has an impact on reducing mortality is very difficult. Many factors during ICU hospitalization determine survival or death, so we tend to evaluate endpoints other than mortality. In the case of MTB, such an endpoint could be a reduction in the need for catecholamines. If we analyze the studies to date for such an endpoint, it appears that the effect of MTB in this regard has been favorable.
The problem in interpreting the results of previous studies is that MTB therapy was mainly based on improving macrocirculatory parameters. Meanwhile, correcting hypotension for the improvement of MAP>65 mmHg recommended by the guidelines does not necessarily translate into improved microcirculation and improved survival if the microcirculation has already been irreversibly damaged.
We look forward to the invention of therapies that will affect the microcirculation. We already have tools such as CytoCam, and perhaps further studies should analyze whether patients in whom there is a positive microcirculatory response to the use of MTB have a better prognosis. This will be a much more personalized approach.
Other minor comments have been taken into account and corrected.
We hope that our work enriched by your comments will be a valuable addition to the data on the use of MTB in refractory vasoplegic shock.
Best regards,
Authors
Reviewer 2 Report
Authors reported a successful use of methylene blue (MTB) in catecholamine-resistant shock and short literature review.
Although this case report is potentially interesting, several issues arise.
Abstract: Authors should show the useful mechanism of MTB in catecholamine-resistant shock.
This manuscript should be revised by native English speaker.
Figure 1 including upper column should be revised. Authors should show the effect of MTB clearly.
Illustration for the effect of MTB on catecholamine-resistant shock may be helpful for the reader.
Was the patient treated with antibiotics.
Abbreviations such as APACHE II, SAPS and SOFA should be explained.
This manuscript should be revised by native English speaker.
Author Response
Thank you for taking the time to adjudicate our manuscript and for your important comments.
“Abstract: Authors should show the useful mechanism of MTB in catecholamine-resistant shock.”
We have supplemented the summary with a brief description of the mechanism of MTB.
“Figure 1 including upper column should be revised. Authors should show the effect of MTB clearly.”
In Figure 1, we have clearly marked the period of MTB administration. In fact, it was not highlighted enough before.
“Illustration for the effect of MTB on catecholamine-resistant shock may be helpful for the reader.”
Thank you for this proposal. We have created a simple graphic showing the mechanism of MTB at the cellular level.
“Was the patient treated with antibiotics.”
Yes, this information is included in the text. The treatment used empirically was meropenem and linezolid.
“Abbreviations such as APACHE II, SAPS and SOFA should be explained.”
Abbreviations have been clarified as suggested by the reviewer.
We hope that with your help our article is now clearer. Thank you again for taking the time to review our work.
Best regards,
authors
Round 2
Reviewer 1 Report
The authors properly accessed my remarks. I have no further comments.
Reviewer 2 Report
Authors fully responded on my comments.
I have no further comments.